

# A geostatistical data-assimilation technique for enhancing macro-scale rainfall-runoff simulations

Alessio Pugliese[1], Simone Persiano[1], Stefano Bagli[2], Paolo Mazzoli[2], Juraj Parajka[3], Berit Arheimer[4], René Capell[4], Alberto Montanari[1], Günter Blöschl[3], and Attilio Castellarin[1]

[1]Department DICAM, Univeristy of Bologna, Bologna, Italy
[2]GECOsistema srl, Cesena, Italy
[3]Institute for Hydraulic and Water Resources Engineering, TU Wien, Wien, Austria
[4]Swedish Meteorological and Hydrological Institute (SMHI), Norrköping, Sweden

*Correspondence to:* Attilio Castellarin (attilio.castellarin@unibo.it)

**Abstract.** Our study develops and tests a geostatistical technique for locally enhancing macro-scale rainfall-runoff simulations on the basis of observed streamflow data that were not used in calibration. We consider Tyrol (Austria and Italy) and two different types of daily streamflow data: macro-scale rainfall-runoff simulations at 11 prediction nodes and observations at 46 gauged catchments. The technique consists of three main steps: (1) period-of-record flow-duration curves (FDCs) are geostatistically predicted at target ungauged basins, for which macro-scale model runs are available; (2) residuals between geostatistically predicted FDCs and FDCs constructed from simulated streamflow series are computed; (3) the relationship between duration and residuals is used for enhancing simulated time series at target basins. We apply the technique in cross-validation to 11 gauged catchments, for which simulated and observed streamflow series are available over the period 1980-2010. Our results show that (1) the procedure can significantly enhance macro-scale simulations (regional NSE increases from nearly zero to $\approx 0.7$ in our study area) and (2) improvements are significant for low gauging network densities (i.e. 1 gauge per 2000 km$^2$).

## 1 Introduction

The steady increase in computational capabilities together with the expanding accessibility of regional and global datasets (e.g. soil properties, land-cover, morphology, climate characteristics, satellite-based gridded precipitation, etc.), trigger the development of regional/continental- and global-scale hydrological models (Archfield et al., 2015), hereafter referred to as macro-scale models.

During the last decade, several of these macroscale models have become operational and thus continuously provide data automatically for decision-making. For instance, the distributed rainfall-runoff-routing model LISFLOOD (De Roo et al., 2000) provides daily forecast for operational warning services through the systems of EFAS (Pappenberger et al., 2013) and GLOFAS (Alfieri et al., 2013); the LARSIM models (Haag and Luce, 2008) are used operationally for simulating streamflow at large areas in southern Germany, Luxembourg, Austria, Switzerland, and the eastern part of France; the WATFLOOD, developed at the University of Waterloo is used operationally in Canada (Kouwen et al., 1993); and the S-HYPE model (Strömqvist





et al., 2012) is running operationally for flood or drought forecasting and water quality assessments for the Swedish landmass, providing high resolution information to authorities and citizens (Hjerdt et al., 2011).

Other macro-scale models are used for off-line water assessments and research purposes. For instance, the global WaterGAP Global Hydrological model (Alcamo et al., 2003) assists in water accounting; the SAFRAN-ISBA-MODCOU model (Habets et al., 2008) has been applied over the entire French territory to combine a meteorological analysis system, a land surface model, and a hydrogeological model; the PGB-IPH model (Pontes et al., 2017) has been applied to many South American basins; and the SWIM model (Krysanova et al., 1998) couples water balance simulations with water quality for small to mid-size watersheds, i.e. regional meso-scale.

The macro-scale hydrological models are getting more and more popular due to three main reasons: (1) they can provide users with a large-scale representation of hydrological behavior, a fundamental information for effectively addressing several water resources planning and management problems (e.g. surface water availability assessment, instream water quality studies, ecohydrological studies, etc.); (2) they can be used to compute a variety of hydrological signatures everywhere along the stream network at the resolution of the model; (3) model outputs in some cases are open-access and freely distributed, their regional runs represent a wealth of information for addressing the problem of hydrological predictions in data scarce regions of the world (Pechlivanidis and Arheimer, 2015; Donnelly et al., 2016; Beck et al., 2016). Accurate regional hydrological simulations undoubtedly foster and support the implementation of improved large-scale and trans-boundary policies for water resources system management and flood-risk mitigation or climate change adaptation (de Paiva et al., 2013; Sampson et al., 2015; Falter et al., 2016; Arheimer et al., 2017).

However, improved accuracy in terms of average regional performance does not necessarily imply homogeneous improve-ments in local performance. In fact, due to the difficulties to perform local calibrations and validations of macro-scale models over the entire modelled regions, local performance can be rather diverse (see e.g., de Paiva et al., 2013; Donnelly et al., 2016). Factors controlling the heterogeneity of local performance may be various, for instance: the quality of macro-scale input data, local water management, representativeness of model structure chosen, the influence of local geophysical and micro-climatic factors, etc.

There is a recognized and noteworthy value of readily available and easy accessible simulated daily streamflow series for scarcely gauged, or ungauged, areas of the world to enhance awareness and decision-making (e.g., Arheimer et al., 2011; Hjerdt et al., 2011; Strömbäck et al., 2013). Nevertheless, the harmonization and enhancement of local performances of macro-scale models is still a scientific challenge that is worth addressing in operational hydrology, and which raises different research questions, such as:

– How to deal with locally biased simulations?

– Can we assimilate additional data to improve model performance without re-calibration?

– Is there a minimum gauging network density that makes the post-modelling data-assimilation viable and effective?

Recent literature shows the significant potential of kriging-based techniques for performing regional prediction of streamflow indices in ungauged locations (Skøien et al., 2006; Castiglioni et al., 2011; Pugliese et al., 2014). Among such techniques,



Topological kriging, or Top-kriging (see Skøien et al., 2006) has shown high prediction accuracy and excellent adaptability to a variety of water-related applications, such as prediction of low-flow indices (Castiglioni et al., 2009), interpolation of river temperatures (Laaha et al., 2013), estimation of flood quantiles (Archfield et al., 2013), regionalization of flow-duration curves (Castellarin, 2014; Pugliese et al., 2014, 2016), estimation of daily runoff in ungauged basins (Parajka et al., 2015) and

reconstruction of historical daily streamflow series (Farmer, 2016).

Our study aims at developing and testing a geostatistical data-assimilation procedure for better agreement between locally observed streamflow and model results from macro-scale rainfall-runoff models. The procedure employs Top-kriging for geostatistically interpolating empirical period-of-record flow-duration curves (FDCs) along the stream network available at gauged basins. Interpolated FDCs are assimilated into simulated daily streamflow series at ungauged stream-network nodes, enhancing

the local accuracy of simulated daily streamflow series. We test our method by improving European HYPE-model simulations (E-HYPE, Donnelly et al., 2016), which provides approximately 30-year of simulated daily streamflows freely and openly accessible for 35 408 (mean catchment size of 215 km$^2$) prediction nodes in Europe (see http://hypeweb.smhi.se/europehype/time-series/). We address the Tyrolean region, as this area gives particularly poor simulations in the HYPE version 2.1 and thus would benefit from statistical enhancement of results. For the geostatistical interpolation, we use a group of 46 gauged

catchments obtained from Austrian and Italian water services, and not used when setting up E-HYPE. With the observed streamflows we construct and interpolate empirical FDCs. Then, we assess the value and potential of assimilating this streamflow information into E-HYPE simulated series. In particular, (1) we cross-validate the proposed data-assimilation procedure for 11 E-HYPE prediction nodes located nearby an existing streamgauge, and (2) we assess the enhancement of simulated series resulting from the geostatistical data-assimilation under different hypotheses on the spatial density of the streamgauging

network. Section 2 presents the proposed procedure in detail, while Sections 3 and 4 illustrate E-HYPE modelling and the study area, respectively. Section 5 details the system of cross-validation and sensitivity analyses we adopted for assessing the procedure and present the results. The last two sections report discussion and conclusions.

## 2   Geostatistical streamflow-data assimilation

### 2.1   Geostatistical interpolation of empirical flow-duration curves (TNDTK)

Top-kriging (or topological kriging) is a powerful geostatistical procedure developed by Skøien et al. (2006) for the prediction of hydrological variables. Like all kriging approaches, Top-kriging produces predictions of hydrological variables at ungauged sites with a linear combination of the empirical information collected at neighbouring gauging stations. Through this method, the unknown value of the streamflow index of interest at prediction location $x_0$, $Z(x_0)$, can be estimated as a weighted average of the variable measured within the neighbourhood:

$$Z(x_0) = \sum_{i=1}^{n} \lambda_i Z(x_i) \tag{1}$$



where $\lambda_i$ is the kriging weight for the empirical value $Z(x_i)$ at location $x_i$, and $n$ is the number of neighbouring stations used for interpolation. Kriging weights $\lambda_i$ can be found by solving the typical ordinary kriging linear system (2a) with the constraint of unbiased estimation (2b):

$$\sum_{j=1}^{n} \gamma_{i,j}\lambda_j + \theta = \gamma_{0,i} \qquad i = 1,\ldots,n \tag{2a}$$

$$\sum_{j=1}^{n} \lambda_j = 1 \tag{2b}$$

where $\theta$ is the Lagrange parameter and $\gamma_{i,j}$ is the semi-variance between catchment $i$ and $j$ (Isaaks and Srivastava, 1990). The semi-variance, or variogram, represents the spatial variability of the regionalised variable $Z$. Unique from any other method of kriging, Top-kriging considers the variable defined over a non-zero support $S$, the catchment drainage area (Cressie, 1993; Skøien et al., 2006). The kriging system of equations (2) remains the same, but the semi-variances between the measurements need to be obtained by regularization, i.e. smoothing the point variogram over the support area. The point variogram can then be back-calculated by fitting aggregated variogram values to the sample variogram (Skøien et al., 2006). Pugliese et al. (2014) proposed a method for using Top-kriging to predict FDCs at ungauged locations that they termed Total Negative Deviation Top-kriging (TNDTK). The authors reduce the dimensionality of the problem by seeking a unique index of site-specific FDCs. Unlike other regional approaches (e.g. regional regression of streamflow quantiles, see e.g. Castellarin et al., 2013) the Top-kriging-based interpolates the entire curve, therefore ensuring its monotonicity (see e.g. Pugliese et al., 2014; Castellarin, 2014). This is accomplished by first standardising the empirical FDCs at site $x$, $\Psi(x,d)$, for some reference value, $Q^*(x)$, to yield a dimensionless FDC:

$$\psi(x,d) = \frac{\Psi(x,d)}{Q^*(x)}, \tag{3}$$

where $d$ denotes a specific duration. Pugliese et al. (2014) identified an overall point index that effectively summarizes the entire curve. This index, which the authors termed total negative deviation (TND), is derived by integrating the area between the lower limb of the FDC and the reference streamflow value $Q^*$ (see Fig. 1). Empirical TND values are computed as:

$$TND(x) = \sum_{i=1}^{m} |q_i(x) - 1|\delta_i, \tag{4}$$

where $q_i = Q_i/Q^*$ represents the $i$-th empirical dimensionless quantile standardised for the selected reference value $Q^*$, $\delta_i$ is half of the frequency interval between the $(i+1)$-th and $(i-1)$-th quantile and the summation involves only the $m$ standardised quantiles lower than 1. The range of the summation, $m$, in Eq. (4) is a function of the maximum duration $d^*$, which is itself a function of that sample with minimum length across gauged sites in the study region. Having calculated empirical TNDs, Pugliese et al. (2014) propose using the TNDs as a regionalised variable to develop site-specific weighting schemes. The




same weights, derived through the solution of the linear kriging system (2), are used for a batch prediction of the continuous, dimensionless FDC for the ungauged site $x_0$:

$$\hat{\psi}(x_0, d) = \sum_{i=1}^{n} \lambda_i \psi(x_i, d) \qquad \forall d \in (0, 1), \tag{5}$$

where $\lambda_i$, with $i = 1, \ldots, n$, are the weights resulting from the kriging interpolation of TNDs; $\psi(x_i, d)$ is the dimensionless, empirical FDC at the donor site $x_i$ and $\hat{\psi}(x_0, d)$ is the predicted dimensionless FDC. It is worth highlighting that the computation of the linear kriging system (2) depends on $n$, the number of neighbouring sites on which to base the spatial interpolation, a fact that will be explored below.

If a reliable model for predicting $Q^*$ at the ungauged site $x_0$ can be developed, the prediction of the dimensional FDC, $\hat{\Psi}(x_0, d)$, is obtained as:

$$\hat{\Psi}(x_0, d) = \hat{Q}^*(x_0) \hat{\psi}(x_0, d) \qquad \forall d \in (0, 1), \tag{6}$$

where $Q^*(x_0)$ is the prediction of $Q^*$ at the ungauged site $x_0$ and $\hat{\psi}(x_0, d)$ has the same meaning as in (5).

## 2.2 New algorithm for geostatistical assimilation of local streamflow data

Following the approach proposed by Smakhtin and Masse (2000), we presents a novel procedure for predicting the model residuals that may be associated with macro-scale rainfall-runoff model simulations (e.g. from LISFLOOD, HYPE, PGB-IPH, etc., see Introduction). This method relies on a regional prediction of the long-term flow-duration curve (FDC) in the same site where these simulations are available.

For instance, let $\Psi(x_0, d)$ be the "true" unknown FDC for a given catchment $x_0$ and $\hat{\Psi}_{SIM}(x_0, d)$ be its prediction constructed on the basis of the daily streamflows simulated through the macro-scale model. We can assume that a general relationship between the two curves exists and reads:

$$\Psi(x_0, d) = \hat{\Psi}_{SIM}(x_0, d) + \varepsilon(x_0, d) \qquad d \in (0, 1), \tag{7}$$

where $\varepsilon(x_0, d)$ are the model residuals defined over the duration domain $d$, which we may term residual-duration curve ($\varepsilon$DC). Evidently, the "true" residual-duration curve is unknown at ungauged basins, nevertheless one can estimate such a curve on the basis of geostatistically interpolated flow-duration curves $\hat{\Psi}_{TNDTK}(x_0, d)$ introduced in Sect. 2.1,

$$\hat{\varepsilon}(x_0, d) = \hat{\Psi}_{TNDTK}(x_0, d) - \hat{\Psi}_{SIM}(x_0, d) \qquad d \in (0, 1). \tag{8}$$

The estimated residual-duration curve obtained from the regional prediction of the long-term flow duration curve can then be used for assimilating local streamflow information into the simulated daily streamflow series. The procedure is sketched in Fig. 2: (1) given a simulated streamflow series (red line in the top-right) select a specific day $t$, and the corresponding discharge $Q(t)$; (2) retrieve the duration $d$ associated with $Q(t)$ from the flow-duration curve constructed from simulated data (red line in



the top-left quadrant); (3) read the estimated residual $\hat{\varepsilon}(t)$ off of the predicted residual-duration curve (blue line in the bottom-left quadrant); and (4) assimilate the residual into the simulated series as $Q(t) + \hat{\varepsilon}(t)$. The iteration of the algorithm through all time steps leads to an enhanced simulated series (blue line in the top-right quadrant).

This new assimilation procedure share some analogies with a technique called "quantile mapping", used in the context of bias corrections for Global Climate Model predictions (see e.g. Komma et al., 2007). The procedure we propose in this context, though, is a rather general tool that can be applied to e.g. any macro-scale rainfall-runoff model for locally enhancing long simulated streamflow series without the need to re-run computationally intensive simulations, provided that the model itself is behavioural and validated on the basis of streamflow data that were not available for model calibration. The performance of the assimilation procedure depends on a variety of drivers, e.g. the quality of the simulated streamflows, which can be severely impacted by the local quality of input data even for a behavioural and well calibrated model, the accuracy of the chosen regional model for predicting FDCs (we refer to TNDTK herein, but there are other viable options, see e.g. Castellarin et al., 2013; Castellarin, 2014), the quality of locally available streamflow data and the density of the streamgauging network. This latter element is specifically investigated in Sect. 5.3.

## 3   Pan-European rainfall-runoff simulation: E-HYPE

The HYdrological Prediction for the Environment (HYPE) model is a hydrological model for small-scale and large-scale assessments of water resources and water quality, developed at the Swedish Meteorological and Hydrological Institute (SMHI) during 2005–2007 (Lindström et al., 2010). The European application, E-HYPE, has been proved to be a powerful tool for water resources managers and practitioners, addressing nutrient concentration in river flow as well as water forecasts on short or seasonal time scale. It is also widely used to estimate snow storage and accumulated TWh of water inflow to hydropower dams and in climate change impact analysis (Donnelly et al., 2017). The website Hypeweb (http://hypeweb.smhi.se) provides visualisation and free downloading of 30 years of continuous and consistent daily streamflow simulations across the European river-network at rather fine scale (i.e. the average size of elementary catchments is equal to 215 km$^2$) as well as forecasts and climate change impact analysis.

The HYPE model is open access and can be downloaded with documentation and model set-up guidelines from the model website (http://hypecode.smhi.se/). It simulates water flow and substances on their way from precipitation through soil, river and lakes to the river outlet (Lindström et al., 2010). River-basins are divided into subbasins, which in turn are divided into classes (the finest calculation units) depending on land use, soil type and elevation (Fig. 3). The soil is modelled as several layers, which may have different thickness for each class. In E-HYPE, each subbasin can have up to some 40 soil and land-use classes, which are lumped within the subbasins, while the watercourses are routed through the river network. The model parameters can be associated with land use (e.g. evapotranspiration), soil type (e.g. water content in soil), or be common for the whole catchment or a region with geophysical similarities (Hundecha et al., 2016). This way of coupling the parameters with geographic information makes the model better suited for simulations in ungauged catchments.



## 4 Study area

We focus on a large alpine region located in Tyrol (Italy, Austria and, for small portion only, Switzerland), for which the E-HYPE model show particularly poor results. Our analyses consider two types of data, observed daily streamflows and E-HYPE simulated daily streamflows, representing different catchments (Fig. 4). This alpine area is particularly suitable for

hydro-power generation, indeed the presence of dams along the stream network might significantly alter the streamflow regime downstream producing a significant alteration of the natural flow conditions. E-HYPE only simulates the dams present in the global database of GranD (Lehner et al., 2011), which might not be representative for hydropower production at the local scale (Arheimer et al., 2017). Therefore, we removed from the initial group of gauged catchments all basins for which the streamflow regime is highly or significantly altered by upstream dams. Table 1 reports the main characteristics of streamflow regimes for

the set of 46 gauged basins and the 11 selected E-HYPE prediction nodes. We selected only those E-HYPE prediction nodes located within Tyrol, which were the closest to one of the available streamgauges. In terms of selection criteria, we selected the E-HYPE catchments that showed limited differences in terms of (1) drainage areas (<14%) and (2) distance between catchment centroids (<15km). These criteria resulted in 11 E-HYPE prediction nodes that are evenly distributed in the study region (see red lines in Fig. 4). We addressed the limited differences existing between drainage areas of E-HYPE and gauged basins by

adopting the drainage-area ratio technique (DAR, see e.g. Farmer and Vogel, 2013), that is by rescaling daily streamflows according to drainage areas of the corresponding catchment. Such method assumes the same unit daily streamflow for any pair of hydrologically similar catchments $i$ and $j$, which reads,

$$\frac{Q_i(t)}{A_i} = \frac{Q_j(t)}{A_j}, \tag{9}$$

where $Q_i(t)$ represents the daily streamflow at day $t$ for catchment $i$ and $Q_j(t)$ is the daily streamflow at day $t$ for catchment

$j$. In our application, $i$ and $j$ could correspond to any given pair (streamgauge, E-HYPE prediction node), and $A_i$ and $A_j$ the corresponding drainage areas.

## 5 Cross-validated predictions: description of resampling procedures and results of the analyses

### 5.1 Cross-validation of the FDC geostatistical interpolator (TNDTK)

We assessed the accuracy of the geostatistical predictor of FDCs (i.e. TNDTK, see Sect. 2.1) in cross-validation over the entire

study area by focusing on the 46 gauged catchments. We chose the mean annual flow (MAF), computed as the average flow of recorded historical streamflow series, as the reference value $Q^*$ (see details in Sect. 2.1). TNDTK operates by first applying Top-kriging to empirical TND values (see Sect. 2.1), which we performed by calculating binned sample variogram first, and then by modelling binned empirical data with a 5-parameter "modified" exponential theoretical variogram (a combination of exponential and a fractal model, see details in Skøien et al., 2006). The fitted theoretical point variogram, and its five parameters

were obtained through the weighted least squares (WLS) regression method from Cressie (1993) by fitting simultaneously all regularised binned variograms that were computed for various area classes (see Skøien, 2014). Recent applications of TNDTK





indicate $n = 6$ as an optimal number of neighbouring donor stations, thus we chose the same value for this case study as well (see details in Pugliese et al., 2014, 2016). Then TNDTK uses the kriging weights obtained for predicting TND values for interpolating the dimensionless FDCs at the location of interest as the weighted average of dimensionless empirical FDCs constructed from the $n = 6$ neighbouring gauged sites (see Eq. (5), in which $\lambda_i$, with $i = 1, \ldots, n$ and $n = 6$, are the kriging

weights).

We adopted a leave-one-out cross-validation (LOOCV) procedure (see e.g. Pugliese et al., 2014, 2016), to test the accuracy and uncertainty associated with FDCs predictions in the study area. This simulates the ungauged conditions at each and every gauged site in the study area by (1) removing it in turn from the dataset and (2) referring to the $n = 6$ neighbouring gauges for predicting its dimensionless FDC. Given that the geostatistical assimilation procedure uses dimensional FDCs, we also tested

the suitability of standard Topkriging for predicting MAF at ungauged locations in the study area, still through a LOOCV procedure (general validity of Topkriging for predicting mean annual flows is also described in Blöschl et al., 2013). For MAF interpolation, we adopted the same settings used for predicting TND values (i.e. a 5-parameter "modified" exponential theoretical variogram and $n = 6$ neighbouring sites).

We then used cross-validated dimensionless FDCs and MAF predictions at each and every gauging station in the study area

to obtain cross-validated predictions of dimensional FDCs for each measuring node through Eq. (6).

We assessed TNDTK performance by means of Nash-Sutcliffe Efficiency (Nash and Sutcliffe, 1970) computed for log-transformed streamflows (LNSE). The application of the geostatistical method TNDTK through an LOOCV procedure reveals a good agreement between empirical values and predictions as shown in Fig. 5 and 6. Specifically, Fig. 5 reports empirical ($x$-axis) against geostatistically predicted ($y$-axis) MAF values as well as LNSE obtained in cross-validation (i.e. 0.96). The

left panel of Fig. 6 shows a scatter diagram between observed ($x$-axis) and predicted streamflows ($y$-axis) from FDCs. This good agreement is confirmed also by the distributions of at-site LNSE values (see box-plots in the right panel of Fig. 6); median LNSEs is equal to 0.97, while mean LNSE is c.a. 0.90. The good performance obtained in cross-validation legitimate the use of TNDTK for predicting FDCs in the study area at the 11 E-HYPE prediction nodes of interest, for which TNDTK delivers very high performances, with LNSE values above 0.97 (see the spatial distribution of efficiency values in the left panel of Fig.

10).

## 5.2 Cross-validation of the geostatistical assimilation procedure (GAE-HYPE)

We applied the proposed assimilation procedure as outlined in Sect. 2.2, focusing on 11 pairs of catchments, i.e. each pair includes one of the E-HYPE catchments depicted in Fig. 4 and its corresponding gauged catchment. We first assessed the efficiency of the procedure in leave-one-out cross-validation. We predicted the FDC associated with each E-HYPE catchment

by using TNDTK and by neglecting the hydrological information coming from the closest (i.e. immediately upstream or downstream) gauged catchment, therefore assuming that no streamflow information is available near the E-HYPE prediction node. The work-flow of the validation algorithm is as follows:





- we select one (E-HYPE prediction node, streamgauge) pair, among $n_{pair} = 11$ possible pairs, let us term it pair $i,j$, where $i$ stands for E-HYPE prediction node and $j$ stands for the corresponding streamgauge; we drop the daily streamflow series observed at streamgauge $j$ from the set of observed series;

- we interpolate FDC at E-HYPE prediction node $i$ by using TNDTK as illustrated in Sect. 2.1 and focusing on the remaining $n_{gauges} - 1$ gauged sites, where $n_{gauges} = 46 - 1$ is total number of streamgauges in the study region minus streamguge $j$;

- we apply the assimilation procedure outlined in Sect. 2.2 and depicted in Fig. 2 to the streamflow series simulated for the E-HYPE prediction node $i$;

- we compare the original E-HYPE daily streamflow series and the geostatistically enhanced one at prediction node $i$ with the daily streamflow series observed at streamgauge $j$ times the corresponding area ratio $A_i/A_j$ (i.e. $A_i$ is the drainage area of E-HYPE prediction node $i$, $A_j$ is the drainage area streamgauge $j$);

- we repeat all previous steps for each one of the remaining $n_{pair-1} = 10$ pairs.

It is worth highlighting here that either TNDTK FDCs or E-HYPE FDCs have been resampled to 20 non-equally-spaced points across duration (see details in Pugliese et al., 2014), as a result, the produced $\hat{\varepsilon}$DC reflect the same sampling scheme of the curves. Nevertheless, the procedure does not foresee any restriction to the resolution of the resampled curve, allowing for a finer resampling scheme in other analyses.

Figure 7 reports the results obtained by applying the aforementioned cross-validation algorithm. Circles in the left panel represent the cumulative absolute error $\delta$ (for details about this metric, see Ganora et al., 2009) computed for each catchment pair $i$-$j$, between empirical FDCs vs. predicted FDCs for either E-HYPE ($\delta_{EHYPE}$ on the $y$ axis) or TNDTK ($\delta_{TNDTK}$ on the $x$ axis) predictions. This figure clearly shows that for 9 out of 11 target sites the geostatistical method TNDTK outperforms E-HYPE in predicting FDCs. Moreover, one of the two sites for which E-HYPE outperforms TNDTK shows nearly the same performance as TNDTK (i.e. the circle is very close to the 1:1 line), while the other one, i.e. site 3675-9001070 highlighted with a black dot in the figure, is associated with the worst performance of TNDTK relative to E-HYPE (see also Sect. 6).

Right panel of Fig. 7 reports estimated residual duration curves ($\hat{\varepsilon}$DCs) for the selected sites. For the sake of representation, we report standardised residuals in the $y$ axis, i.e. residuals divided by the corresponding streamflow quantiles predicted via TNDTK; we referred to TNDTK quantiles for standardization since the real empirical FDC is supposed to be unknown (see cross-validation algorithm illustrated above). Overall, $\hat{\varepsilon}$DCs show negative values for lower durations and positive values for higher durations (see also Fig. 9). This means that, in Tyrol, E-HYPE tends to overestimate streamflow in wet periods as well as to underestimate streamflows in drier ones relative to the geostatistically predicted FDCs (i.e. TNDTK). We eventually used the $\hat{\varepsilon}$DC curves, which are estimates of E-HYPE residuals, to assimilate locally available streamflow data into E-HYPE simulated series as illustrated in Fig. 2, obtaining what we term GAE-HYPE simulations.

Representativeness of simulations (i.e. E-HYPE and GAE-HYPE simulated daily streamflows) is assesses through Nash-Sutcliffe Efficiency of log-flows (LNSE) computed by referring to the whole E-HYPE simulation period. We found remarkable



improvements obtained with the proposed data-assimilation procedure (Fig. 8). Indeed, one can notice a substantial enhance-ment of LNSE values of GAE-HYPE simulations relative to the original E-HYPE ones, which in the best case increases from -0.462 to 0.594 (catchment pair IDs: 201236 - 9608296) and in the worst case from 0.527 to 0.690 (catchment pair IDs: 3675 - 9001070). The median at-site LNSE value increases from 0.045 to 0.685, which ultimately underline the benefits introduced

with the proposed method. Figure 8, also, illustrates the impact of geostatistical data-assimilation for the two E-HYPE predic-tion nodes mentioned above (i.e. the one characterized by the best improvements in terms of overall LNSE value, and the one associated with the most limited improvement). In both cases the advantages associated with the data-assimilation procedure are evident.

Finally, looking at the spatial distribution of LNSE values across the 11 selected prediction nodes within the study area

depicted in Fig. 10, it is clear how the proposed enhancement strategy benefits from an unbiased estimations of FDCs. In fact, TNDTK shows homogeneous and rather good performance for predicting FDCs (left panel of Fig. 10); also, middle and right panels of Fig. 10 reveal that the enhancement capabilities of the assimilation procedure are lower for those catchments where E-HYPE performs better (see elementary catchments filled in yellow to green in Fig. 10), whereas the assimilation procedure proves to be very powerful when E-HYPE performs worse (see elementary catchment coloured in orange to red), bringing

efficiencies from negative to positive values in all cases (from green to blue).

### 5.3   Streamgauging network density and effectiveness of geostatistical data-assimilation

Since the proposed geostatistical data-assimilation procedure relies upon the local availability of streamgauges records, under-standing to what extent the performance of the assimilation method are driven by gauging network density is of paramount importance. Therefore, we performed a sensitivity analysis and assessed the degree of enhancement of simulated daily stream-

flow sequences associated with different scenarios of streamflow data availability, repeating for each scenario the procedure described in Sect. 5.2.

We randomly discarded some of the gauges available over the study area (i.e. 46 in total) and varied the total number of available stations continuously from 7 (lowest density) to 46 (highest density) gauges (i.e. 40 iterations in total), that means approximately from 4 to 29 gauges per $10\,000\,\mathrm{km}^2$. We repeated these 40 iterations for each and every E-HYPE prediction node,

11 in total, by using exactly the same settings illustrated in detail in Sect. 5.1 and 2.1 for implementing the data-assimilation procedure (e.g. 6 donor sites, exponential-fractal theoretical variogram, MAF is the index-flow discharge, etc.). We assessed the efficiency of the data-assimilation procedure through the following metric,

$$LNSE_{ratio} = \frac{LNSE_{GAE-EHYPE} - LNSE_{EHYPE}}{1 - LNSE_{EHYPE}}, \tag{10}$$

where $LNSE_x$ is the log-flows Nash-Sutcliffe efficiency for simulated daily streamflows, and $x$ could either refer to E-HYPE

(original) or GAE-HYPE (after data-assimilation).

$LNSE_{ratio}$ quantifies the degree of enhancement of GAE-HYPE relative to E-HYPE standardized by the maximum possible improvement (i.e. $1 - LNSE_{GAE-EHYPE}$). An $LNSE_{ratio}$ close to zero means no significant enhancement (detriment of



original sequences in case of negative values), whereas an $LNSE_{ratio}$ close to 100% indicates that no further enhancement is possible.

Figure 11 shows $LNSE_{ratio}$ values we obtained under different scenarios of streamflow data availability and displays a clear pattern, which confirms the improvement of the degree of enhancement associated with an increasing gauge density.

Figure 11 shows how the degree of enhancement flattens out in cases in which there are more than 20 gauges available per $10\,000\,\text{km}^2$. Another noteworthy feature of Fig. 11 is the remarkable enhancement that can be obtained for the scenario with the lowest gauge density (i.e. 7 gauging stations, minus 1 because the sensitivity analysis is performed in leave-one-out cross validation, is equal to 6, which is exactly the number of donor sites we chose for the implementation of Top-kriging).

## 6 Discussion

This new geostatistical procedure enables practitioners and water resources managers and planners to profit from the wealth of hydrological information, by adjusting open data products with local observations. We enhanced the streamflow series simulated by macro- and continental scale rainfall-runoff models at ungauged prediction nodes by assimilating streamflow observations, which are locally available in the region of interest, without having to redo the original hydrological model calculations. This is a recurrent condition since local streamflow data are released under different license terms and policies:

some of them could be public and open-access, while some other might not be openly and freely accessible by the broad public. The E-HYPE model obviously lack storage capacity in the Tyrol region and the proposed approach to enhance the results should be seen as temporary until a new model version accounting for this is available. We do not propose the procedure as a general fix for structurally unsuitable (or non-behavioural, see Beven and Binley, 1992) models, which have been proved to be unfit for either the area of interest or the water-problem at hand. For a more sustainable solution, we suggest to use

another model structure or re-calibrate the model, instead of post-processing the output. However, this procedure makes sense for making a first assessment of water issues in regions were information is otherwise missing, but only macro-scale models are readily available.

Our study shows for Tyrol that indeed it is possible to significantly enhance rainfall-runoff simulations resulting from macro-scale, regional or continental hydrological models by geostatistically assimilating (geographically sparse) streamflow obser-

25 vations (see e.g. Fig. 8, 9 and 10); provided that available streamflow series are long enough to obtain a good empirical approximation of the long-term flow-duration curve (FDC) for the site of interest (i.e. 5-10 years, see Castellarin et al., 2013).

One of the main advantages of the proposed method is that the end-user can get enhanced streamflow simulations without any further model calibration or refinement. Even though one could argue that when additional streamflow data become available at neighbouring gauges, it should be used for improving the performance of the model at the site of interest, calibrating and

30 validating macro-scale and regional model could be a time-consuming and computationally demanding task. The proposed procedure, instead, is neither computational nor data intensive, and is implemented only using observed streamflow data and a GIS vector layer with catchment boundaries (see e.g. Fig. 4). The application requires the identification of a suitable regional model for predicting FDC in ungauged basis (see e.g. Fig. 6). However, it has advantages, such as: (a) a regional model can





be a very informative and useful tool for water resources managers and planners; and (b) the subsequent advantages obtained from the data-assimilation procedure is transferred downstream in the entire regional river-network (see Fig. 11).

One important limitation of the proposed method is that, once a target prediction node is considered, any given simulated streamflow value is associated with a single duration, which corresponds to a particular estimated residual, that will be used

in turn for correcting the streamflow value itself (see Fig.s 2 and 9). This algorithm cannot possibly account for seasonal (or interannual) modifications in the hydrological behaviour of the catchment. Indeed, as shown in time series comparison of Fig. 9, when the geostatistical prediction of FDCs is unreliable, then neither improvements nor detriments are delivered by the proposed procedure (i.e. the procedure fails to correctly capture high-flow regimes, see e.g. the catchment pair 3675-9001070 in Fig. 7 and 9), propagating this bias throughout the whole simulated series.

Finally, designing a theoretical framework that combines statistical data-driven approaches with deterministic process-driven ones is seen by many as the correct way for tackling the 'Prediction in Ungauged Basins (PUB)' problem and further advancing the scientific research in this area (see e.g. Di Prinzio et al., 2011). We believe that our geostatistical data-assimilation procedure for macro-scale hydrological models is one example in this direction. Future analyses will focus on the relaxation of the main limitation of the approach (i.e. the incorporation of seasonal patterns in data-assimilation procedure) and on the extension of

its applicability to anthropogenically altered streamflow regimes.

## 7 Conclusions

This research work focuses on the development of an innovative method for enhancing streamflow series simulated by macro-, continental-, and global-scale rainfall-runoff models by means of a geostatistical prediction of model residuals. We focus on Tyrol as study region and E-HYPE (European - HYdrological Predictions for the Environment, from the Swedish Meteo-

20 rological and Hydrological Institute, SMHI) as a macro-scale hydrological model, respectively; nevertheless, the geostatistical data-assimilation procedure is general and can be applied to simulated streamflow-series coming from other macro-scale rainfall-runoff models. The proposed data-assimilation procedure utilises streamflow-data that are locally available for the area of interest, which were not considered in the implementation of the macro-scale hydrological model; it (1) adopts Top-kriging for regionally interpolating empirical period-of-record flow-duration curves (FDCs) that can be constructed from locally avail-

25 able streamflow data; (2) constructs residual-duration relationships at any prediction node in the study region where simulated streamflow series are available by comparing FDCs resulting from geostatistical interpolation (Topkriging) and rainfall-runoff simulation (E-HYPE); (3) uses the error-duration curve to enhance macro-scale simulated streamflows.

The cross validation tests of the proposed approach with different scenarios of streamflow data availability, shows the significant advantages of geostatistical data-assimilation even for very low stream-gauging network densities (i.e. c.a. 1 gauge per

30 2000 km$^2$). It can become a standalone numerical tool to be used for enhancing results from macro-scale models anywhere along the stream network of a given region. Potential applications are envisaged for a variety of water resources management and planning problems that require accurate streamflow series (e.g. regional assessment of hydropower potential, habitat suitability studies, surface water allocation, etc.). Future analyses will address the main limitation of the proposed geostatis-





tical data-assimilation procedure, aiming at incorporating observed seasonal and inter-annual variations of the hydrological behaviour of the study region into the geostatistical regionalization of model residuals.

*Code and data availability.* The analysis was carried out in the Virtual Water Science Lab developed within the FP7 funded research project SWITCH-ON (grant agreement no. 603587). We invite the interested reader to explore the experiment protocol here: http://dl-ng005.xtr.

5   deltares.nl/view/462/

*Acknowledgements.* The analyses presented in the study are the main outcomes of the international experiment Geostatistical Enhancement of European Hydrological Predictions (GEEHP), which was performed in the Virtual Water-Science Lab developed within the European Commission FP7 funded research project SWITCH-ON (Sharing Water-related Information to Tackle Changes in the Hydrosphere – for Operational Needs, grant agreement no. 603587). The overall aim of the project is to promote data sharing to exploit open data sources. The

10   study also contributes to developing the framework of the "Panta Rhei" Research Initiative of the International Association of Hydrological Sciences (IAHS). The majority of figures included in the study were produced by the use of free and open source softwares (i.e. Quantum GIS Geographic Information System - Open Source Geospatial Foundation Project, http://qgis.osgeo.org, and the R Project for Statistical Computing, https://www.R-project.org/)



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



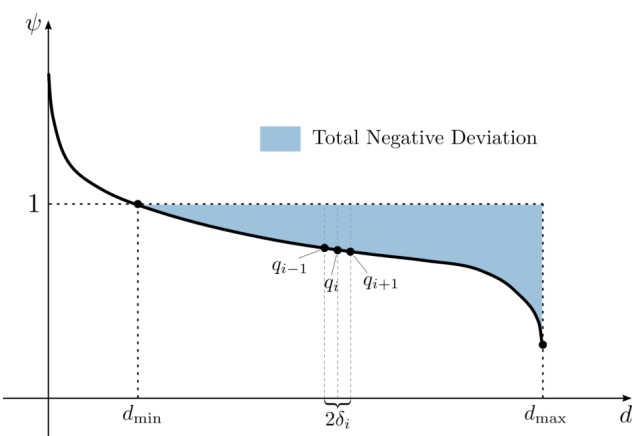

**Figure 1.** A sketch of the Total Negative Deviation (TND).

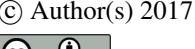



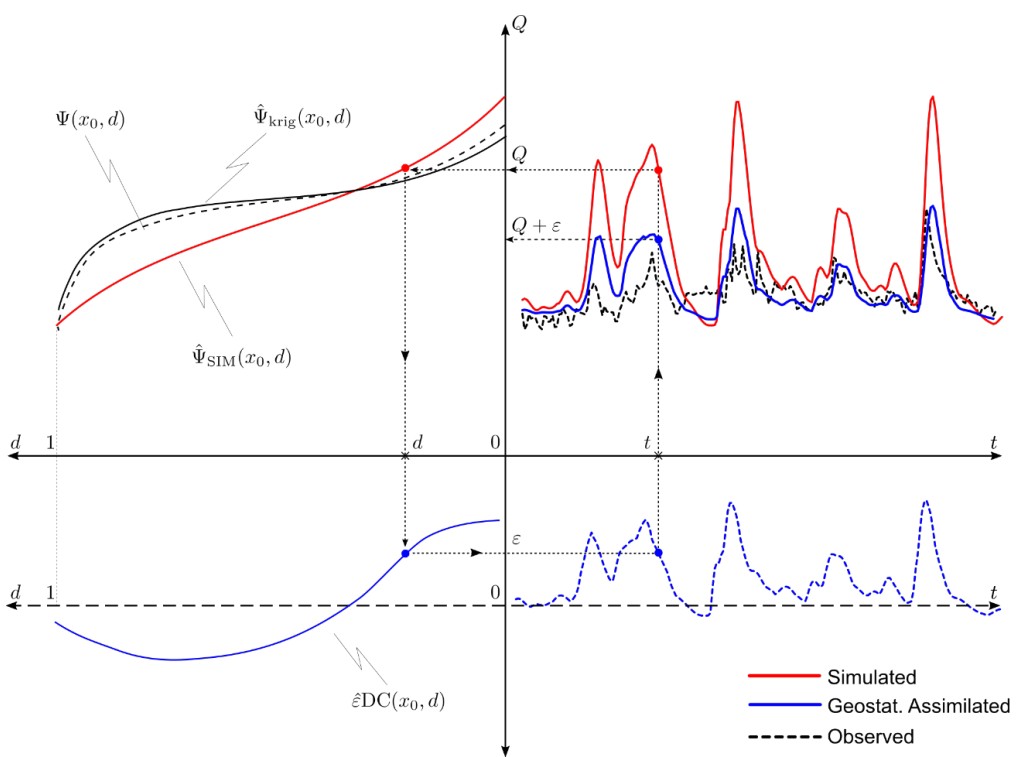

**Figure 2.** Illustration of the proposed data-assimilation procedure for a given simulated time series. Top-right panel: real streamflow series (unknown, since the basin is ungauged, black dashed line); macro-scale model simulation (red solid line); geostatistically-enhanced streamflow series (blue solid line). Top-left panel: FDCs predictions obtained from simulated streamflows (red solid line) and via geostatistical interpolation (black solid line); real (unknown) FDC (black dashed line). Bottom-left panel: estimated residual-duration curve (blue solid line) computed as the difference between the two predicted FDCs in the top-let panel. Bottom-right panel: time series of residuals (blue dashed line).



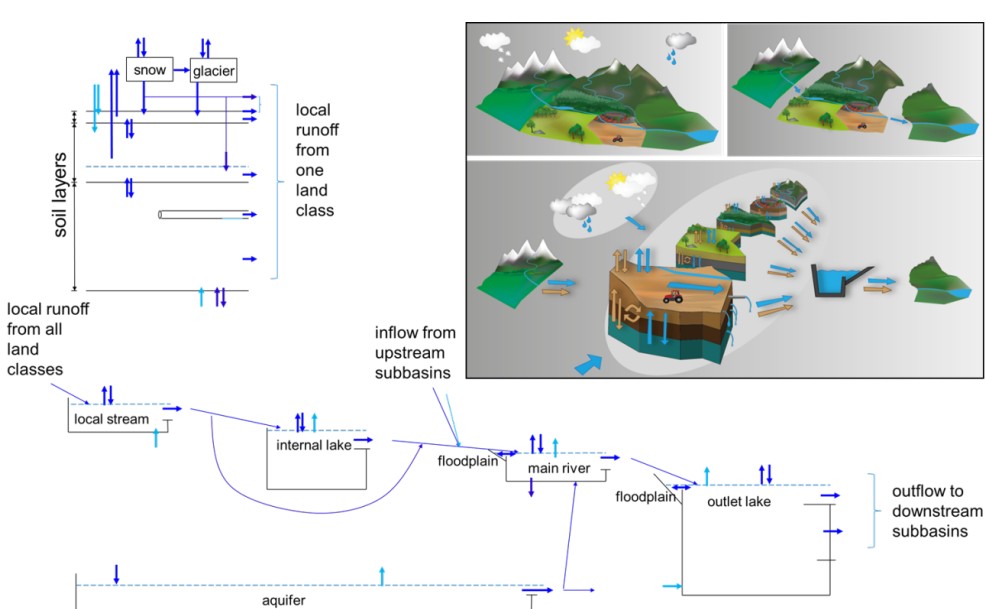

**Figure 3.** Schematic concept of the HYPE model (all equations are available at http://hypecode.smhi.se/).




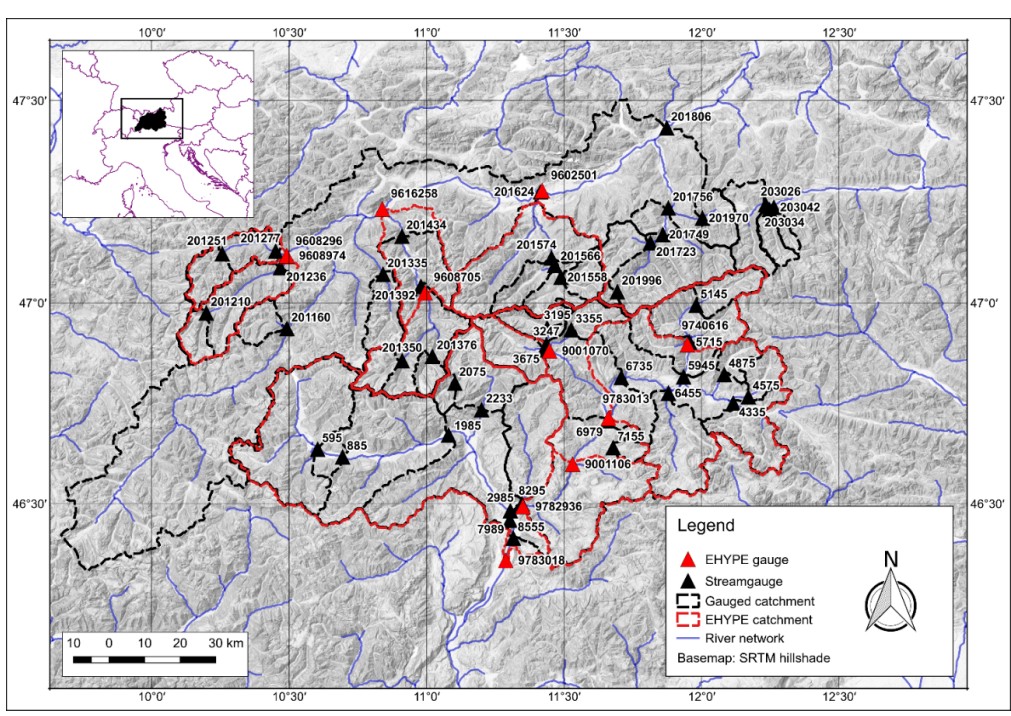

**Figure 4.** Study area: Tyrol. Catchment boundaries for 11 E-HYPE prediction nodes (red) and 46 stream gauges (black).



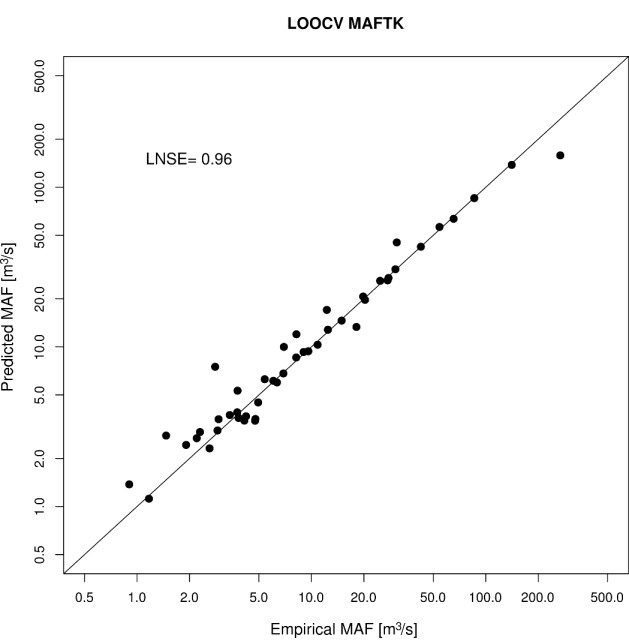

**Figure 5.** Top-kriging predictions of mean annual flow (MAF) in cross-validation mode.





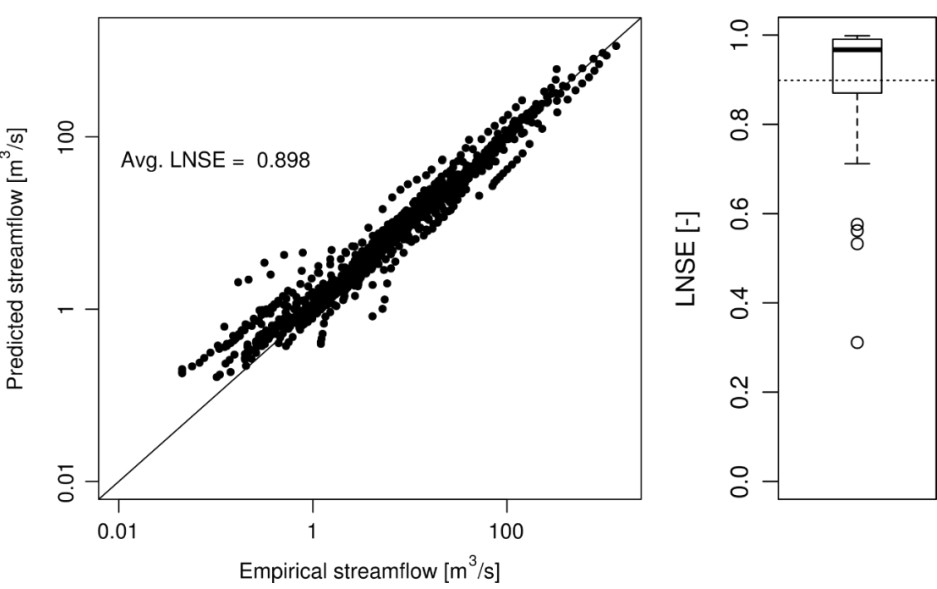

**Figure 6.** Left panel: scatter diagrams of empirical ($x$-axes) vs. predicted ($y$-axis) streamflows. Right panel: box-plot representation of at-site LNSE values, summarizing the 1st, 2nd (median) and 3rd quartiles along with whiskers extending to the most extreme non-outlying data point (outliers are highlighted as circles and lay at more than 1.5 times the interquartile-range from the nearest quartile); average at-site LNSE value is reported in the left panel and illustrated as a dashed line in the right one.





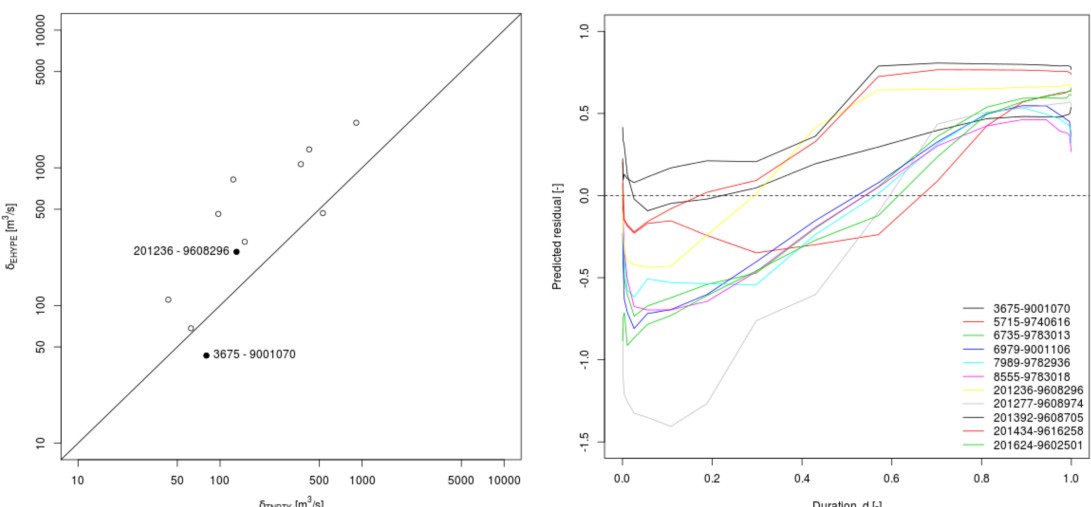

**Figure 7.** 11 E-HYPE prediction nodes: left panel, comparison between TNDTK ($x$-axis) and E-HYPE ($y$-axis) in terms of distances $\delta$s between empirical and predicted FDCs; the 1:1 line represents equivalent performance for TNDTK and E-HYPE; right panel, standardizaed residual-duration curves computed as illustrated in eq. (8) (TNDTK streamflow quantiles are used for standardization).





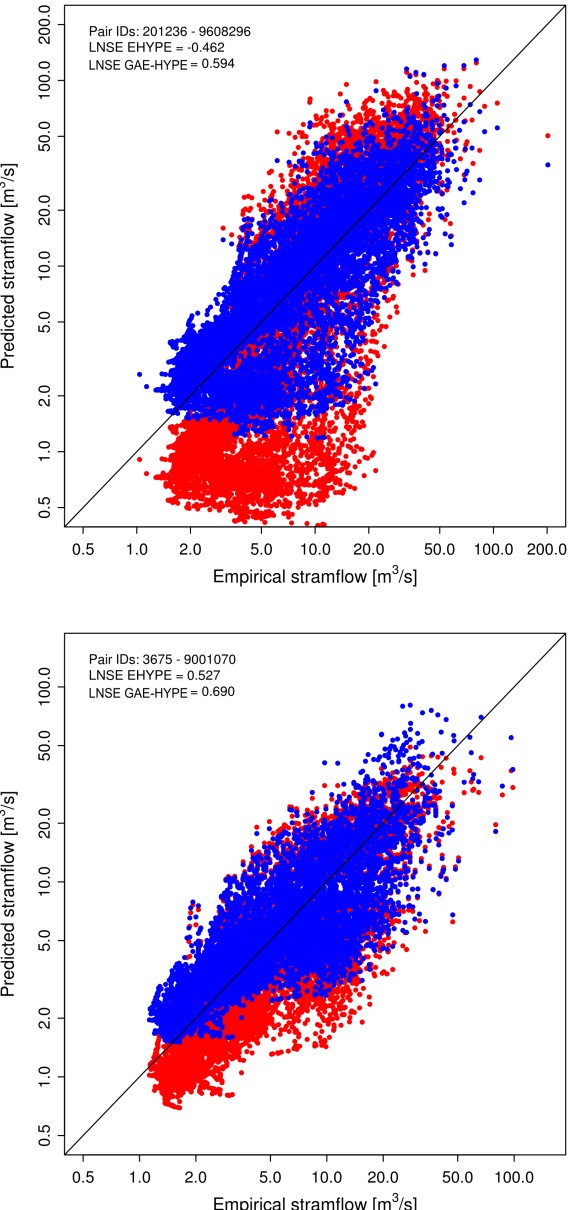

**Figure 8.** Scatter diagrams of empirical vs. simulated daily streamflows for either E-HYPE (red dots) and GAE-HYPE (blue dots) for two representative sites, showing the cases in which the data-assimilation procedure respectively produced the largest (upper panel) and smallest (lower panel) degree of enhancement for the study area, respectively.





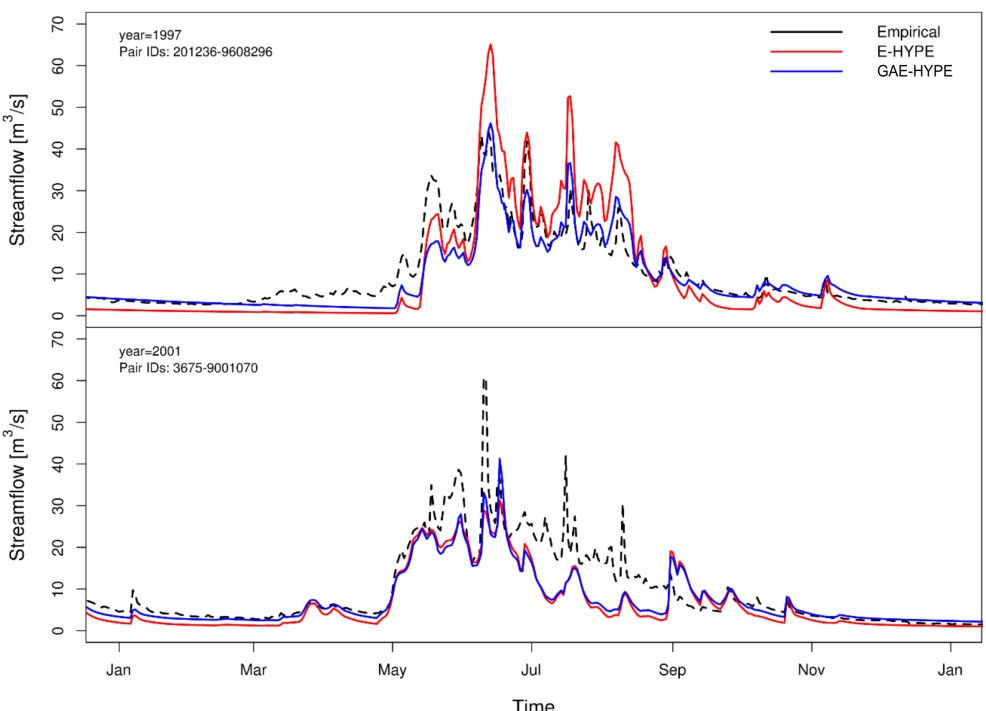

**Figure 9.** Examples of comparison between observed streamflow series (back dashed lines) and simulated daily streamflows via E-HYPE (red solid lines) and GAE-HYPE (blue solid lines) for two representative sites and a given year, showing two cases for which the geostatistical assimilation procedure resulted in sizeable (top panel) and limited (bottom panel) improvements, respectively.





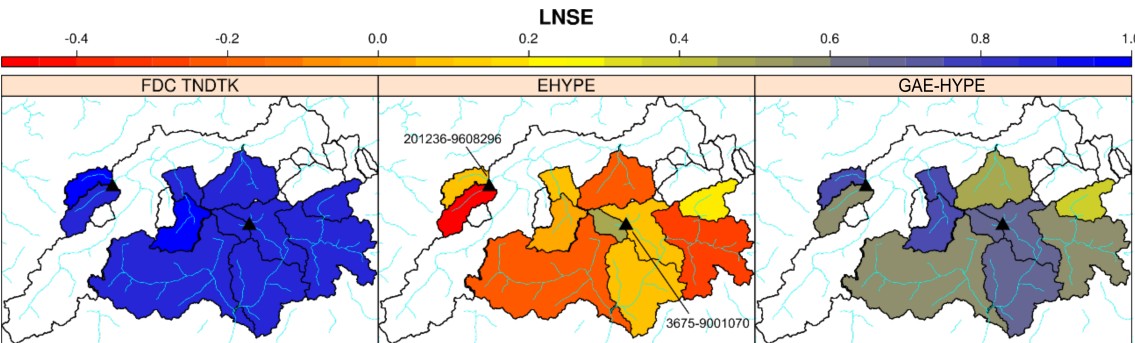

**Figure 10.** Spatial distribution of Nash-Sutcliffe Efficiency computed for log-transformed streamflows (LNSE) at the 11 E-HYPE prediction nodes considered in the study; geostatistically predicted flow-duration curves (FDC TNDTK, left); predicted daily streamflow time series (E-HYPE, centre, and GAE-HYPE, right, respectively); the locations of the two sites considered in Figure 9 are highlighted with black triangles.





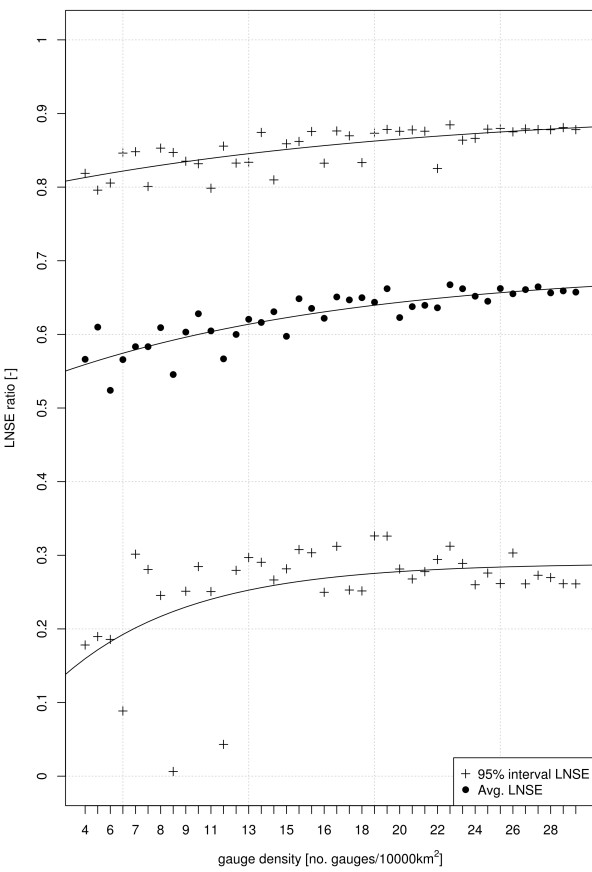

**Figure 11.** LNSE ratio (see eq. (19)) as a function of streamgauge availability: black dots represent the average of 11 LNSE ratio values, while crosses indicate their 95% confidence interval.





**Table 1.** Study catchments: streamflow properties standardised by drainage area [m$^3$/s/km$^2$] for either gauged catchments or E-HYPE catchments, mean annual flow (qMAF), 50% and 95% streamflow quantiles (q50 and q95, respectively).

|  | Gauged catchments (46) | | | E-HYPE catchments (11) | | |
|---|---|---|---|---|---|---|
|  | qMAF | q50 | q95 | qMAF | q50 | q95 |
| Min. | 0.0147 | 0.0078 | 0.0023 | 0.0261 | 0.0057 | 0.0009 |
| 75th percentile | 0.0205 | 0.0158 | 0.0046 | 0.0276 | 0.0117 | 0.0022 |
| Median | 0.0309 | 0.0188 | 0.007 | 0.0294 | 0.0169 | 0.0032 |
| Mean | 0.0315 | 0.0195 | 0.0066 | 0.034 | 0.0168 | 0.0028 |
| 25th percentile | 0.0369 | 0.0221 | 0.008 | 0.0351 | 0.0223 | 0.0035 |
| Max. | 0.0588 | 0.043 | 0.0116 | 0.0622 | 0.0275 | 0.0047 |

**Table 2.** Nash-Sutcliffe Efficiencies computed on log-transformed daily streamflows for E-HYPE and GAE-HYPE: median values for the 11 prediction nodes considered in the study; smallest enhancement (IDs 3675 - 9001070), largest enhancement (IDs 201236 - 9608296).

| LNSE |  |  | E-HYPE | GAE-HYPE |
|---|---|---|---|---|
| **Median** |  |  | 0.045 | 0.685 |
| **Pair** | GAUGE ID | E-HYPE ID |  |  |
| Smallest enhancement | 3675 | 9001070 | 0.527 | 0.69 |
| Largest enhancement | 201236 | 9608296 | -0.462 | 0.594 |