# Peer review of "A geostatistical data-assimilation technique for enhancing macro-scale rainfall-runoff simulations"

_Hydrology and Earth System Sciences, 2017_

## Referee Comment (RC1) · W.H. Farmer (Referee) · 17 Oct 2017

The authors have presented a very well-prepared manuscript exploring the value of post-processing or "bias-correcting" daily streamflow simulations with independently derived regional information. By using an independently developed flow duration curve, the authors present an approach to customize macro-scale models to local conditions. The approach is very valuable. The manuscript is well-written and thorough. Below, I will provide some minor comments, but I see no impediment to swift publication.

Was the performance of GAE-HYPE always better than the performance of E-HYPE? At the top of page 10, the authors cite the best and worst improvements. Were there

any sites that showed decreases in performance? If not, would you expect universal improvement in other regions?

Was the degree of change in LNSE between GAE-HYPE and E-HYPE a function of the accuracy with which the TNDTK FDC was produced? That is, if the TNDTK FDC were poorly produced, it seems like the improvement in LNSE might be reduced or reversed. Some exploration of this might be useful.

What method of interpolation was used to map the residuals to simulated streamflows? On line 13 of page 9, it is stated that the TNDTK FDC is resampled to 20 points. So, if a simulated streamflow (E-HYPE) produced a duration that did not fall at one of the 20 resampled points, how was the residual estimated? That is, how was the eDC resolved between these 20 points?

Does the simulated FDC resulting from the GAE-HYPE series match the TNDTK FDC? It seems as if it could (should?), by the nature of the method. This suggests that this method could also be considered as a re-scaling of the simulated streamflow distribution. Essentially, this means that the volumes from E-HYPE are discarded while the sequencing of E-HYPE (durations, relative values) is retained. I do not see anything wrong with this, but wonder if it is another way to think about the procedure. If accurate, does this way of thinking provide any further insight?

What was the significance of the changes in LNSE? Firstly, equation 10 can be simplified as the fractional change in root-mean-squared error of logarithms. This, can, of course, be interpreted as a percent. (Line 1 of page 11 uses percent, but the figure does not.) More importantly, the LNSE values could be compared in a pairwise test to determine if the improvement in LNSE is statistically significant (Wilcoxon). I imagine it is, but demonstrating this would provide stronger evidence.

How were streamflow values of zero handled? The authors measure performance in terms of the LNSE. What was the frequency of zeros? How were there logarithms taken?

[Figure]

How many bins were used to discretize the variograms? Binning is described on line 27 of page 7, but it might be worthwhile to be explicit.

Editorial: On line 16 of page 1 the authors use "macro-scale" while line 17 uses "macroscale". Both are used throughout the manuscript; select one.

Editorial: The figures seem to be out of order. Fig. 9 is mentioned after Fig. 7 and before Fig. 8.

Finally, I sincerely thank the authors for their work and their well-written manuscript. It was easy to read, which made it easy to consider its technical merit. I look forward to their considered response and hope to have ignited some useful thought and discussion. If you would like any clarification on my comments, I strongly encourage you to reach out to me.

Thanks, William Farmer

---

## Referee Comment (RC2) · Anonymous Referee #2 · 1 Nov 2017

1. General Comments: This paper "A geostatistical data-assimilation technique for enhancing macro-scale rainfall-runoff simulations" develops a geostatistical method for enhancing streamflow simulation performance of large-scale rainfall-runoff models. The proposed method has proved to be effective for Tyrol area and shows great potential for basins with few gauges or even ungauged basins.

2. Some major comments: (1) Organizations of Section 5 seem to be not logical. Following "Section 4: study area", some new method (e.g. LOOCV) and new metric (e.g. LNSE) are introduced in Section 5, and then followed by the findings and discussions. This may confuse readers as they may fail to grasp the intention and the core of this

paper. It had better reorganize the manuscript in 'Method-Results' order. (2) The concept of total negative deviation (TND) is very essential to this paper's research. So, it is recommended that TND be explained more clearly with both words and figure. Particularly, Figure 1 should be interpreted in detail, for instance, what does the symbol '1' exactly stand for? (3) Why choose the mean annual flow (MAF) as the reference value Q* in applying equation (3)? Does this mean that you didn't consider the flow above MAF when applying "TNDTK", according to the definition of TND by the shaded area in Figure 1? If this is the case, more should be elaborated on this. (4) Page 8 line 28. It stated that "each pair includes one of the E-HYPE catchments depicted in Fig. 4 and its corresponding gauged catchment." How to determine the corresponding gauged catchment for a certain E-HYPE catchment? By Equation 4 or other method? Please explain in detail.

3. Minor Comments: (1) I notice that Equation 1 takes the symbol 'i' as indicator of catchments, while in Equation 4 it represents the quantiles of qi. This should be avoided. Please use a different symbol. (2) In Figure 9, the black dashed line indicates the observed streamflow series, which is not in line with its legend, where the black solid line is plotted. This is a minor mistake that could have been avoided. (3) In Figure 10, the meaning of the black triangles is interpreted in the title. This is not a good idea. Please use legends instead. (4) What does 'TNDTK' mean in the title of Section 2.1? DO NOT use abbreviation before it is defined.

---

## Referee Comment (RC3) · Anonymous Referee #3 · 15 Nov 2017

(1) The symbol should confirm, such as equation (6); (2) This author exploring a technique for the daily streamflow simulation post processing, whether the name "data-assimilation technique" is ok or another name is better ; (3) This method used the information from local information (such as rainfall, landuse and topography) and neighbor watershed data (observation runoff data for FDC), Can we get the effect of local/ neighbor information on different part of runoff simulation (such as peak flow and baseflow) combined this technique; (4) Whether the method can be applied in real time forecasting, and hope the author give us some perspective; (5) I tried to do the example, but I cannot download the Observed daily streamflow series. http://www.water-switch-on.eu/sip-webclient/sip-beta/#/resource/12072.

---

## Author Comment (AC2) · 15 Jan 2018

*Referee comment:*

General Comments: This paper "A geostatistical data-assimilation technique for enhancing macro-scale rainfall-runoff simulations" develops a geostatistical method for enhancing streamflow simulation performance of large-scale rainfall-runoff models. The proposed method has proved to be effective for Tyrol area and shows great potential for basins with few gauges or even ungauged basins.

[Figure]

*Authors reply:*

We thankfully acknowledge reviewer's useful comments, which will help us to improve the presentation of the study.

*Referee comment:*

Some major comments: (1) Organizations of Section 5 seem to be not logical. Following "Section 4: study area", some new method (e.g. LOOCV) and new metric (e.g. LNSE) are introduced in Section 5, and then followed by the findings and discussions. This may confuse readers as they may fail to grasp the intention and the core of this paper. It had better reorganize the manuscript in 'Method-Results' order.

*Authors reply:*

Although we see reviewer's comment, we did not introduce LOOCV and LNSE earlier in the text as these are a common cross-validation procedure (see e.g. Kroll and Song, 2013; Salinas et al., 2013; Wan Jaafar et al., 2011; Srinivas et al., 2008) and a widely used metric (see e.g. Farmer, 2016; Castellarin, 2014). Yet, we do agree that this may generate some confusion.

**ACTION**: We decided to introduce a new section in the manuscript, in between "Study area" and "Results" sections, titled "4 Assessment of the geostatistical assimilation algorithm" in which we present the structure of the analysis, the cross-validation strategies and the performance index. This section will report the following subsections:

4.1 Structure of the analysis

4.2 Cross-validation strategy
4.3 Performance indices

Nevertheless, before changing the sectioning, and thus the whole structure of the paper, we would like to wait for editor decision.

*Referee comment:*

(2) The concept of total negative deviation (TND) is very essential to this paper's research. So, it is recommended that TND be explained more clearly with both words and figure. Particularly, Figure 1 should be interpreted in detail, for instance, what does the symbol '1' exactly stand for?

*Authors reply:*

We are grateful to the reviewer for highlighting this lack of clarity behind the idea of TND. We will add more information about this novel metric in the revised version of the paper. TND is conceived to mimic the slope of a standardised FDC, where the standardisation is an arbitrary reference value (e.g. mean annual flow; see also Pugliese et al., 2014, for other standardisation methods). Therefore, "1" on the $y$-axis in Fig. 1 represents the equality between a given streamflow record and the reference value.

**ACTION**: We will add this sentence in P4 L25 : "[...] The equality between a given streamflow value and the reference value $Q^*$ is represented by an horizontal dashed line in Fig. 1, i.e. the threshold given by the equation $q/Q = 1$. [...]"

*Referee comment:*

(3) Why choose the mean annual flow (MAF) as the reference value in applying equation (3)? Does this mean that you didn't consider the flow above MAF when applying

"TNDTK", according to the definition of TND by the shaded area in Figure 1? If this is the case, more should be elaborated on this.

*Authors reply:*

We understand that this paper neither explains carefully the idea behind the TND, nor it presents a thorough assessment of its reliability as regional metric summarising FDC's, however we think that this is out of the scope of this paper. Indeed, such assessments have been already carried out in other two independent research studies, where we proposed the TNDTK method and we contrasted its performances against other regional models, such as regional regression and statistical models, which are known to be the state-of-the-art for regional FDC predictions in ungauged basins (see Pugliese et al., 2016, 2014).

**ACTION**: We will better clarify this in the revised manuscript, providing an interested reader with indications on original literature sources.

*Referee comment:*

(4) Page 8 line 28. It stated that "each pair includes one of the E-HYPE catchments depicted in Fig. 4 and its corresponding gauged catchment." How to determine the corresponding gauged catchment for a certain E-HYPE catchment? By Equation 4 or other method? Please explain in detail.

*Authors reply:*

The selection criteria of the 11 sites used for comparing the proposed technique with EHYPE simulations have been explained in section 4 Study area (which will be Section

5 in the revised manuscript).

**ACTION**: We will change the sentences on P7 L10

"[. . .] We selected only those E-HYPE prediction nodes located within Tyrol, which were the closest to one of the available streamgauges. In terms of selection criteria, we selected the E-HYPE catchments that showed limited differences in terms of (1) drainage areas ($<$14%) and (2) distance between catchment centroids ($<$15km). These criteria resulted in 11 E-HYPE prediction nodes that are evenly distributed in the study region (see red lines in Fig. 4). [. . .]" with

"[. . .] Among all E-HYPE prediction nodes available in Tyrol we selected only those whose catchments were the closest to gauged ones, i.e. differences in terms of drainage areas $<$14% and distance between catchment centroids $<$15km. These criteria resulted in the selection of 11 E-HYPE prediction nodes that are evenly distributed in the study region (see red lines in Fig. 4). [. . .]". Finally, it is worth noting that Eq. (4) does not deal with catchment selection, but with empirical TND computation only.

*Referee comment:*

Minor Comments: (1) I notice that Equation 1 takes the symbol $i$ as indicator of catchments, while in Equation 4 it represents the quantiles of qi. This should be avoided. Please use a different symbol. (2) In Figure 9, the black dashed line indicates the observed streamflow series, which is not in line with its legend, where the black solid line is plotted. This is a minor mistake that could have been avoided. (3) In Figure 10, the meaning of the black triangles is interpreted in the title. This is not a good idea. Please use legends instead. (4) What does 'TNDTK' mean in the title of Section 2.1? DO NOT use abbreviation before it is defined.

*Authors reply:*

(1) The reviewer is right. We will substitute $i$ with $j$ in Eq. (1). (2) Thanks. We will change accordingly. (3) Thanks. We will add a legend for both the black triangles and catchment boundaries. (4) Thanks. We will drop both TNDTK and GAE-HYPE from the title of the sections.

**References**

Castellarin, A., 2014. Regional prediction of flow-duration curves using a three-dimensional kriging. J. Hydrol. 513, 179–191. https://doi.org/10.1016/j.jhydrol.2014.03.050

Komma, J., Reszler, C., Blöschl, G., Haiden, T., 2007. Ensemble prediction of floods – catchment non-linearity and forecast probabilities. Nat Hazards Earth Syst Sci 7, 431–444. https://doi.org/10.5194/nhess-7-431-2007

Kroll, C.N., Song, P., 2013. Impact of multicollinearity on small sample hydrologic regression models. Water Resour. Res. 49, 3756–3769. https://doi.org/10.1002/wrcr.20315

Pugliese, A., Castellarin, A., Brath, A., 2014. Geostatistical prediction of flow–duration curves in an index-flow framework. Hydrol Earth Syst Sci 18, 3801–3816. https://doi.org/10.5194/hess-18-3801-2014

Pugliese, A., Farmer, W.H., Castellarin, A., Archfield, S.A., Vogel, R.M., 2016. Regional flow duration curves: Geostatistical techniques versus multivariate regression. Adv. Water Resour. 96, 11–22. https://doi.org/10.1016/j.advwatres.2016.06.008

Salinas, J.L., Laaha, G., Rogger, M., Parajka, J., Viglione, A., Sivapalan, M., Blöschl, G., 2013. Comparative assessment of predictions in ungauged basins - Part 2: Flood and low flow studies. Hydrol. Earth Syst. Sci. 17, 2637–2652. https://doi.org/10.5194/hess-17-2637-2013

Srinivas, V.V., Tripathi, S., Rao, A.R., Govindaraju, R.S., 2008. Regional flood frequency analysis by combining self-organizing feature map and fuzzy clustering. J. Hydrol. 348, 148–166. https://doi.org/10.1016/j.jhydrol.2007.09.046

Wan Jaafar, W.Z., Liu, J., Han, D., 2011. Input variable selection for median flood regionalization. Water Resour. Res. 47, W07503. https://doi.org/10.1029/2011WR010436

---

## Author Comment (AC3) · 15 Jan 2018

*Referee comment:*

(1) The symbol should confirm, such as equation (6).

*Authors reply:*

Thanks.

[Figure]

**ACTION**: the missing cap over $Q^*(x_0)$ will be added in the revised version.

*Referee comment:*

(2) This author exploring a technique for the daily streamflow simulation post process-ing, whether the name "data-assimilation technique" is ok or another name is better

*Authors reply:*

There are some similarities between our technique and applications in climate mod-elling (see e.g. Komma et al., 2007). We would like to keep the title as it is now.

*Referee comment:*

This method used the information from local information (such as rainfall, landuse and topography) and neighbour watershed data (observation runoff data for FDC), Can we get the effect of local/neighbour information on different part of runoff simulation (such as peak flow and baseflow) combined this technique.

*Authors reply:*

The power of the proposed technique relies on the fact that no further observations than mere discharges are needed for enhancing streamflow simulations. Surely, it would be interesting to investigate how other hydrological features, such as baseflow index or peakflow data, might be assimilated in the method.

**ACTION**: We will underline in the discussion section that future research studies will deal this problem.

*Referee comment:*

Whether the method can be applied in real time forecasting, and hope the author give us some perspective.

*Authors reply:*

In principle it could be used by blending this assimilation technique to e.g. long term forecast, even though the proposed method cannot be applied without any locally observed streamflow series.

**ACTION**: We will add in the discussion section that future analyses will assess the reliability of the method with the final aim to provide better simulations for practitioners at operational level, e.g. applications in civil protection management strategies, climate change trends, safety of river structures, etc.

*Referee comment:*

I tried to do the example, but I cannot download the Observed daily streamflow series. http://www.water-switchon.eu/sip-webclient/sip-beta/#/resource/12072.

*Authors reply:*

(5) Thanks. We will fix the broken link.

**References**

Komma, J., Reszler, C., Blöschl, G., Haiden, T., 2007. Ensemble prediction of floods – catchment non-linearity and forecast probabilities. Nat Hazards Earth Syst Sci 7, 431–444. https://doi.org/10.5194/nhess-7-431-2007

———————————————————

---

## Author Response (AR1)

\* References to lines and pages, coloured in blue in the following, refer to the revised version of the manuscript.

\*\* Tracked changes version of the manuscript is attached at the end of this document.

**❖ Reply to referee W.H. Farmer**

*Referee comment:*

The authors have presented a very well-prepared manuscript exploring the value of post-processing or "bias-correcting" daily streamflow simulations with independently derived regional information. By using an independently developed flow duration curve, the authors present an approach to customize macro-scale models to local conditions. The approach is very valuable. The manuscript is well-written and thorough. Below, I will provide some minor comments, but I see no impediment to swift publication.

*Authors reply:*

We thank Reviewer #1 W. H. Farmer for his valuable and precise comments. We are pleased of the appreciation for the overall study, and we hope that the replies to his comments presented below might resolve the issues arisen.

*Referee comment:*

Was the performance of GAE-HYPE always better than the performance of E-HYPE? At the top of page 10, the authors cite the best and worst improvements. Were there any sites that showed decreases in performance? If not, would you expect universal improvement in other regions?

*Authors reply:*

Yes, we obtained improvements in each one of the 11 pairs, however it is worth recalling that EHYPE was not calibrated in those sites, which results in locally poor performance of EHYPE. Overall, improvements are always to be expected when FDCs are well predicted by the geostatistical interpolator.

ACTION: In the revised version of the manuscript we will underline that all selected pairs show improvements in the performances.

Done. See P12 L8.

*Referee comment:*

Was the degree of change in LNSE between GAE-HYPE and E-HYPE a function of the accuracy with which the TNDTK FDC was produced? That is, if the TNDTK FDC were poorly produced, it seems like the improvement in LNSE might be reduced or reversed. Some exploration of this might be useful.

*Authors reply:*

The reviewer is right, the whole procedure relies on good prediction of regional FDC in the study area. When FDCs are poorly predicted across the area, detriments of the geostatistical assimilation procedure are to be expected. Focusing on the 11 EHYPE prediction nodes, FDCs are actually very well predicted, with an average LNSE of 0.988 in cross- validation (with minimum and maximum values about 0.967 and 0.998, respectively) compared to the average performance in the whole study area, which is 0.898. Therefore, we cannot show here the effects of poor geostatistical interpolation of FDCs on the assimilation procedure.

ACTION: we will specifically highlight in the revised manuscript that biased regional model for predicting FDCs are expected to lead to poor performances of the proposed geostatistical assimilation procedure.

Done. See P6 L14-17.

*Referee comment:*

What method of interpolation was used to map the residuals to simulated streamflows? On line 13 of page 9, it is stated that the TNDTK FDC is resampled to 20 points. So, if a simulated streamflow (E-HYPE) produced a duration that did not fall at one of the 20 resampled points, how was the residual estimated? That is, how was the eDC resolved between these 20 points?

*Authors reply:*

We thank the reviewer for letting us explain further this technical aspect. Observed FDCs are resampled to 20 equally spaced points, in the normal space, i.e. d ~{0.0004; 0.0013; 0.0040; 0.0108; 0.0259; 0.0558; 0.1080; 0.1884; 0.2979; 0.4298; 0.5702; 0.7021; 0.8116; 0.8920; 0.9442; 0.9741; 0.9892; 0.9960; 0.9987; 0.9996}. The extreme values of the resampling scheme depends on the length of the shorter observed streamflow series in the dataset (i.e. 7 yrs), so that, those values extruding from such extremes are excluded in the resampled curves. Each EHYPE simulation covers a period of 30 yrs. with no gap, therefore we adopted the same resampling interval, with 20 points, used for the observed series. As a result, the residual duration curves are defined in the same 20 points of the resampling scheme and a linear interpolation is employed in between each data point. Finally, according to Pugliese et al. (2014), regional interpolation of TNDs, which is the weighting scheme for the curves as well, does not require any resampling scheme beforehand.

ACTION: in the revised version, we will add more information on the resampling strategy we used and we will clearly underline that a finer resolution of the curves could be necessary in specific applications, in which very high (low) duration are of particular interest.

Done. See P8 L21-25, and P13 L14-17.

*Referee comment:*

Does the simulated FDC resulting from the GAE-HYPE series match the TNDTK FDC? It seems as if it could (should?), by the nature of the method. This suggests that this method could also be considered as a re-scaling of the simulated streamflow distribution. Essentially, this means that the volumes from E-HYPE are discarded while the sequencing of E-HYPE (durations, relative values) is retained. I do not see anything wrong with this, but wonder if it is another way to think about the procedure. If accurate, does this way of thinking provide any further insight?

*Authors reply:*

The reviewer rises a good point here.

ACTION: In the revised version of the manuscript we will add two panels in fig. 9 where we show FDCs of either empirical, E-HYPE, or GAE-HYPE next to their streamflow series counterparts, since we believe that the result is worth showing. We will also include in the discussion an analysis resulting from this comparison and possible insights on the aspect highlighted by the reviewer (the proposed geostatistical assimilation procedure discards simulated volumes, while retaining simulated sequencing).

Done. See the revised fig. 9, see also P13 L27-30.

*Referee comment:*

What was the significance of the changes in LNSE? Firstly, equation 10 can be simplified as the fractional change in root-mean-squared error of logarithms. This, can, of course, be interpreted as a percent. (Line 1 of page 11 uses percent, but the figure does not.) More importantly, the LNSE values could be compared in a pairwise test to determine if the improvement in LNSE is statistically significant (Wilcoxon). I imagine it is, but demonstrating this would provide stronger evidence.

*Authors reply:*

We thank the reviewer for these recommendations, which we will add in the revised version of the manuscript. We are aware that the proposed $LNSE_{ratio}$ can be simplified as the fractional change in root-mean-squared error of logarithms, therefore, we cannot see much difference in using one or the other, however, for the sake of consistency, we would prefer to keep this assessment metric in terms of LNSE.

ACTIONS:

1) We will clearly highlight that the proposed index derives from the fractional change in root-mean-squared error of logarithms.
2) We will improve the consistency between notation and units in figures and text.
3) In addition, for each gauge density scenario, we will perform a pairwise test (Wilcoxon test) between LNSE values in order to determine whether improvements are statistically significant in terms of LNSE.

Done:

1) See P9 L26.

2) See e.g. equations subscripts switched from *i* to *j* whenever they refer to streamgauge spatial index.

3) See P9 L27-29 and P12 L25-27.

*Referee comment:*

How were streamflow values of zero handled? The authors measure performance in terms of the LNSE. What was the frequency of zeros? How were there logarithms taken?

*Authors reply:*

Although the alpine climate might be at risk for the presence of zero flows within the series, especially during the winter seasons, we observed no zero flows in the selected 11 sites with either empirical, EHYPE or GAE-HYPE.

ACTION: We will report such information in section 4.

Done. See P10 L22-23.

*Referee comment:*

How many bins were used to discretize the variograms? Binning is described on line 27 of page 7, but it might be worthwhile to be explicit.

*Authors reply:*

We set the number of bins as the default setting of the "rtop" package, which is 1 bin per order of magnitude of drainage area. In our case, areas span over 4 orders of magnitude, therefore we used 3 bins for variogram pairs.

ACTION: We will add this information too.

We double-checked the results and the number of bins used, relative to three orders of magnitude in the drainage areas range, are actually 2. We reported this information at P7 L13-15.

*Referee comment:*

Editorial: On line 16 of page 1 the authors use "macro-scale" while line 17 uses "macroscale". Both are used throughout the manuscript; select one.

*Authors reply:*

Thanks.

ACTION: We will use "macro-scale" and change accordingly.

Done.

*Referee comment:*

Editorial: The figures seem to be out of order. Fig. 9 is mentioned after Fig. 7 and before Fig. 8.

*Authors reply:*

Thanks.

ACTION: We will change accordingly.

Done.

**❖ **Reply to anonymous referee#2**

*Referee comment:*

General Comments: This paper "A geostatistical data-assimilation technique for enhancing macro-scale rainfall-runoff simulations" develops a geostatistical method for enhancing streamflow simulation performance of large-scale rainfall-runoff models. The proposed method has proved to be effective for Tyrol area and shows great potential for basins with few gauges or even ungauged basins.

*Referee comment:*

Some major comments: (1) Organizations of Section 5 seem to be not logical. Following "Section 4: study area", some new method (e.g. LOOCV) and new metric (e.g. LNSE) are introduced in Section 5, and then followed by the findings and discussions. This may confuse readers as they may fail to grasp the intention and the core of this paper. It had better reorganize the manuscript in 'Method-Results' order.

*Author reply:*

Although we see reviewer's comment, we did not introduce LOOCV and LNSE earlier in the text as these are a common cross-validation procedure (see e.g. Kroll and Song, 2013; Salinas et al., 2013; Wan Jaafar et al., 2011; Srinivas et al., 2008) and a widely used metric (see e.g. Farmer, 2016; Castellarin, 2014). Yet, we do agree that this may generate some confusion.

ACTION: We decided to introduce a new section in the manuscript, in between "Study area" and "Results" sections, titled "4 Assessment of the geostatistical assimilation algorithm" in which we present the structure of the analysis, the cross-validation strategies and the performance index. This section will report the following subsections:

4.1 Structure of the analysis

4.2 Cross-validation strategy

4.3 Performance indices

Nevertheless, before changing the sectioning, and thus the whole structure of the paper, we would like wait for editor decision.

The structure has been fully revised. We decided to modify the sectioning by moving the structure proposed above before the description of study area. Please, pay attention to changes applied at sections 2, 3, and 4 whereas the results have been blended to a single section.

*Referee comment*:

(2) The concept of total negative deviation (TND) is very essential to this paper's research. So, it is recommended that TND be explained more clearly with both words and figure. Particularly, Figure 1 should be interpreted in detail, for instance, what does the symbol '1' exactly stand for?

*Authors reply*:

(2) We are grateful to the reviewer for highlighting this lack of clarity behind the idea of TND. We will add more information about this novel metric in the revised version of the paper. TND is conceived to mimic the slope of a standardised FDC, where the standardisation is an arbitrary reference value (e.g. mean annual flow; see also Pugliese et al., 2014, for other standardisation methods). Therefore, "1" on the *y*-axis in Fig. 1 represents the equality between a given streamflow record and the reference value.

ACTION: We will add this sentence in P4 L25 : "[…] The equality between a given streamflow value and the reference value $Q^*$ is represented by an horizontal dashed line in Fig. 1, i.e. the threshold given by the equation $Q/Q^*=1$. […]"

Done. See P4 L25-26.

*Referee comment*:

(3) Why choose the mean annual flow (MAF) as the reference value $Q^*$ in applying equation (3)? Does this mean that you didn't consider the flow above MAF when applying "TNDTK", according to the definition of TND by the shaded area in Figure 1? If this is the case, more should be elaborated on this.

*Authors reply*:

(3) We understand that this paper neither explains carefully the idea behind the TND, nor it presents a thorough assessment of its reliability as regional metric summarising FDC's, however we think that this is out of the scope of this paper. Indeed, such assessments have been already carried out in other two independent research studies, where we proposed the TNDTK method and we contrasted its performances against other regional models, such as regional regression and statistical models, which are known to be the state-of-the-art for regional FDC predictions in ungauged basins (see Pugliese et al., 2016, 2014).

ACTION: We will better clarify this in the revised manuscript, providing an interested reader with indications on original literature sources.

Done. See P5 L12-15.

*Referee comment*:

(4) Page 8 line 28. It stated that "each pair includes one of the E-HYPE catchments depicted in Fig. 4 and its corresponding gauged catchment." How to determine the corresponding gauged catchment for a certain E-HYPE catchment? By Equation 4 or other method? Please explain in detail.

*Authors reply*:

(4) The selection criteria of the 11 sites used for comparing the proposed technique with EHYPE simulations have been explained in section 4 Study area (which will be Section 5 in the revised manuscript).

ACTION: We will change the sentences on P7 L10 "[…] We selected only those E-HYPE prediction nodes located within Tyrol, which were the closest to one of the available streamgauges. In terms of selection criteria, we selected the E-HYPE catchments that showed limited differences in terms of (1) drainage areas (<14%) and (2) distance between catchment centroids (<15km). These criteria resulted in 11 E-HYPE prediction nodes that are evenly distributed in the study region (see red lines in Fig. 4). […]"

with

"[…] Among all E-HYPE prediction nodes available in Tyrol we selected only those whose catchments were the closest to gauged ones, i.e. differences in terms of drainage areas <14% and distance between catchment centroids <15km. These criteria resulted in the selection of 11 E-HYPE prediction nodes that are evenly distributed in the study region (see red lines in Fig. 4). […]".

Finally, it is worth noting that Eq. (4) does not deal with catchment selection, but with empirical TND computation only.

Done. See P10 L12-15.

*Referee comment:*

Minor Comments: (1) I notice that Equation 1 takes the symbol 'i' as indicator of catchments, while in Equation 4 it represents the quantiles of qi. This should be avoided. Please use a different symbol. (2) In Figure 9, the black dashed line indicates the observed streamflow series, which is not in line with its legend, where the black solid line is plotted. This is a minor mistake that could have been avoided. (3) In Figure 10, the meaning of the black triangles is interpreted in the title. This is not a good idea. Please use legends instead. (4) What does 'TNDTK' mean in the title of Section 2.1? DO NOT use abbreviation before it is defined.

*Authors reply*:

(1) The reviewer is right. We will substitute *i* with *j* in Eq. (1).

(2) Thanks. We will change accordingly.

(3) Thanks. We will add a legend for both the black tringles and catchment boundaries.

(4) Thanks. We will drop both TNDTK and GAE-HYPE from the title of the sections.

1) Done. See eq. 1.

2) Done. See fig. 9.

3) Done. See fig. 10.

4) Done.

❖ **Reply to anonymous referee#3**

*Referee comment:*

(1) The symbol should confirm, such as equation (6).

*Authors reply:*

Thanks.

ACTION: the missing cap over $\hat{Q}^*(x_0)$ will be added in the revised version.

Done. See eq. 6.

*Referee comment*:

(2) This author exploring a technique for the daily streamflow simulation post processing, whether the name "dataassimilation technique" is ok or another name is better

*Authors reply*:

There are some similarities between our technique and applications in climate modelling (see e.g. Komma et al., 2007). We would like to keep the title as it is now.

*Referee comment*:

This method used the information from local information (such as rainfall, landuse and topography) and neighbour watershed data (observation runoff data for FDC), Can we get the effect of local/ neighbour information on different part of runoff simulation (such as peak flow and baseflow) combined this technique.

*Authors reply*:

The power of the proposed technique relies on the fact that no further observations than mere discharges are needed for enhancing streamflow simulations. Surely, it would be interesting to investigate how other hydrological features, such as baseflow index or peakflow data, might be assimilated in the method.

ACTION: We will underline in the discussion section that future research studies will deal this problem.

Done. See P14 L24-26.

*Referee comment*:

Whether the method can be applied in real time forecasting, and hope the author give us some perspective.

*Author comment*:

In principle it could be used by blending this assimilation technique to e.g. long term forecast, even though the proposed method cannot be applied without any locally observed streamflow series.

ACTION: We will add in the discussion section that future analyses will assess the reliability of the method with the final aim to provide better simulations for practitioners at operational level, e.g. applications in civil protection management strategies, climate change trends, safety of river structures, etc.

Done. Here we revised the conclusions rather than the discussion section. See P14 L21-24.

*Referee comment*:

I tried to do the example, but I cannot download the Observed daily streamflow series. http://www.water-switchon.eu/sip-webclient/sip-beta/#/resource/12072.

*Authors reply*:

(5) Thanks. We will fix the broken link.

Links fixed. Please, refer to the one in the revised version of the paper.

**References**

[revised manuscript text omitted]

---

## Author Response (AR2)

Reply to reviewer W. H. Farmer

Reviewer comment:

*The authors have satisfactorily responded to all my comments, and I see no impediment to publication. The authors thoroughly document an approach to data-assimilation that can be used to "bias-correct" simulated streamflows. The following are only minor comments.*

*The methods are presented before the data. While this is not unacceptable, I think the manuscript reads more logically when the data are presented first.*

*The acronym HYPE is used in the introduction, before being defined several sections later. Given that this is the underlying model being used, I think it worthwhile to define it upfront.*

*On page four, a paragraph break is needed in line 11.*

*In the results section, modifiers like "good" and "remarkable" seem overly subjective to me. If possible, I would revise to remove the implication of subjectivity by merely stating the magnitude of improvement or quantifying performance directly. Subjectivity, if used, should be reserved for the discussion.*

*On page 11, line 19, I believe "legitimate" should be "legitimizes".*

*In the results section, the figures are presented as Fig 7, 9 and 8. The figures should probably be re-numbered for the order of presentation.*

*Thank you for the opportunity to review this revision. As always, do not hesitate to reach out if you would like to discuss any of my comments further.*

*Thanks again,*
*William Farmer*

Authors reply:

We would like to thankfully acknowledge William H. Farmer for his valuable comments, which undoubtedly helped us in further polishing our presentation. We agree with the reviewer on the fact that presenting methodologies and procedures before the data can sometime result in a less logical structure; however, the current structure of the manuscript is a direct consequence of having addressed the requests that Reviewer#2 raised during the first round of reviews. Furthermore, we believe that presenting methods and procedures beforehand, makes the presentation more general and the methods themselves more directly applicable to different contexts. We now clarify this view at the end of the Introduction. In conclusion, we would prefer to leave the sectioning as it is, even though we are open to a further restructuring, if the Editor deems it necessary.

Aside from the sectioning, we corrected the manuscript following thoroughly all recommendations raised by William Farmer. All these corrections are highlighted in the tracked-changes version of our manuscript, which is appended below.

[revised manuscript text omitted]